# Data-efficient Active Learning for Structured Prediction with Partial Annotation and Self-Training

**Zhisong Zhang, Emma Strubell, Eduard Hovy**
Language Technologies Institute, Carnegie Mellon University
zhisongz@cs.cmu.edu, strubell@cmu.edu, hovy@cmu.edu

## Abstract

In this work we propose a pragmatic method that reduces the annotation cost for structured label spaces using active learning. Our approach leverages partial annotation, which reduces labeling costs for structured outputs by selecting only the most informative sub-structures for annotation. We also utilize self-training to incorporate the current model's automatic predictions as pseudo-labels for unannotated sub-structures. A key challenge in effectively combining partial annotation with self-training to reduce annotation cost is determining which sub-structures to select to label. To address this challenge, we adopt an error estimator to adaptively decide the partial selection ratio according to the current model's capability. In evaluations spanning four structured prediction tasks, we show that our combination of partial annotation and self-training using an adaptive selection ratio reduces annotation cost over strong full annotation baselines under a fair comparison scheme that takes reading time into consideration.

## 1 Introduction

Structured prediction (Smith, 2011) is a fundamental problem in NLP, wherein the label space consists of complex structured outputs with groups of interdependent variables. It covers a wide range of NLP tasks, including sequence labeling, syntactic parsing and information extraction (IE). Modern structured predictors are developed in a data-driven way, by training statistical models with suitable annotated data. Recent developments in neural models and especially pre-trained language models (Peters et al., 2018; Devlin et al., 2019; Liu et al., 2019; Yang et al., 2019) have greatly improved system performance on these tasks. Nevertheless, the success of these models still relies on the availability of sufficient manually annotated data, which is often expensive and time-consuming to obtain.

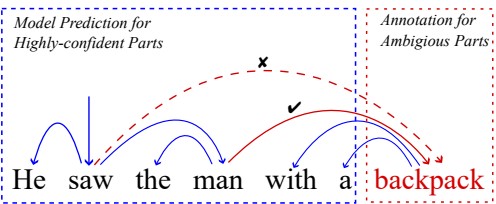

Figure 1: Example partial annotations of a dependency tree. Manual annotation is requested only for the uncertain sub-structures (red), whereas model predictions can be used to annotate the highly-confident edges (blue).

To mitigate such data bottlenecks, active learning (AL), which allows the model to select the most informative data instances to annotate, has been demonstrated to achieve good model accuracy while requiring fewer labels (Settles, 2009). When applying AL to structured prediction, one natural strategy is to perform full annotation (FA) for the output structures, for example, annotating a full sequence of labels or a full syntax tree. Due to its simplicity, FA has been widely adopted in AL approaches for structured prediction tasks (Hwa, 2004; Settles and Craven, 2008; Shen et al., 2018). Nevertheless, a structured object can usually be decomposed into smaller sub-structures having non-uniform difficulty and informativeness. For example, as shown in Figure 1, in a dependency tree, edges such as functional relations are relatively easy to learn, requiring fewer manual annotations, while prepositional attachment links may be more informative and thus more worthwhile to annotate.

The non-uniform distribution of informative sub-structures naturally suggests AL with partial annotation (PA), where the annotation budget can be preserved by only choosing a portion of informative sub-structures to annotate rather than laboriously labeling entire sentence structures. This idea has been explored in previous work, covering typical structured prediction tasks such as sequence labeling (Shen et al., 2004; Marcheggiani and Artières, 2014; Chaudhary et al., 2019; Radmard et al., 2021)

**Algorithm 1** AL Procedure.

**Input:** Seed dataset $\mathcal{L}_0$, dev dataset $\mathcal{D}$, unlabeled pool $\mathcal{U}$, total budget $t$, batch selection size $b$, annotation $strategy$.
**Output:** Final labeled dataset $\mathcal{L}$, trained model $\mathcal{M}$.
1: $\mathcal{L} \leftarrow \mathcal{L}_0$    # Initialize
2: **while** $t > 0$ **do**    # Until out of budget
3:    $\mathcal{M} \leftarrow$ train($\mathcal{L}, \mathcal{U}$)    # Model training
4:    $\mathcal{S} \leftarrow$ sentence-query($\mathcal{M}, \mathcal{U}$)    # Sentence selection
5:    **if** $strategy$ == "partial" **then**
6:        $r \leftarrow$ auto-ratio($\mathcal{S}, \mathcal{D}$)    # Decide adaptive ratio
7:        partial-annotate($\mathcal{S}, r$)    # Partial annotation
8:    **else**
9:        full-annotate($\mathcal{S}$)    # Full annotation
10:    $\mathcal{U} \leftarrow \mathcal{U} - \mathcal{S}$; $\mathcal{L} \leftarrow \mathcal{L} \cup \mathcal{S}$; $t \leftarrow t - b$
11: $\mathcal{M} \leftarrow$ train($\mathcal{L}, \mathcal{U}$)    # Final model training
12: **return** $\mathcal{L}, \mathcal{M}$

and dependency parsing (Sassano and Kurohashi, 2010; Mirroshandel and Nasr, 2011; Flannery and Mori, 2015; Li et al., 2016). Our work follows this direction and investigates the central question in AL with PA of how to decide which sub-structures to select. Most previous work uses a pre-defined fixed selection criterion, such as a threshold or ratio, which may be hard to decide in practice. In this work, we adopt a performance predictor to estimate the error rate of the queried instances and decide the ratio of partial selection accordingly. In this way, our approach can automatically and adaptively adjust the amount of partial selection throughout the AL process.

Another interesting question for AL is how to better leverage unlabeled data. In this work, we investigate a simple semi-supervised method, self-training (Yarowsky, 1995), which adopts the model's automatic predictions on the unlabeled data as extra training signals. Self-training naturally complements AL in the typical pool-based setting where we assume access to a pool of unlabeled data (Settles, 2009). It is particularly compatible with PA-based AL since the un-selected sub-structures are typically also highly-confident under the current model and likely to be predicted correctly without requiring additional annotation. We revisit this idea from previous work (Tomanek and Hahn, 2009; Majidi and Crane, 2013) and investigate its applicability with modern neural models and our adaptive partial selection approach.

We perform a comprehensive empirical investigation on the effectiveness of different AL strategies for typical structured prediction tasks. We perform fair comparisons that account for the hidden cost of reading time by keeping the context size the same for all the strategies in each AL cy-

cle. With evaluations on four benchmark tasks for structured prediction (named entity recognition, dependency parsing, event extraction, and relation extraction), we show that PA can obtain roughly the same benefits as FA with the same reading cost but less sub-structure labeling cost, leading to better data efficiency. We also demonstrate that the adaptive partial selection scheme and self-training play crucial and complementary roles.

## 2 Method

### 2.1 AL for Structured Prediction

We adopt the conventional pool-based AL setting, which iteratively selects and annotates instances from an unlabeled pool. Please refer to Settles (2009) for the basics and details of AL; our main illustration focuses more specifically on applying AL to structured prediction.

Algorithm 1 illustrates the overall AL process. We focus on sentence-level tasks. In FA, each sentence is annotated with a full structured object (for example, a label sequence or a syntax tree). In PA, annotation granularity is at the sub-structure level (for example, a sub-sequence of labels or a partial tree). We adopt a two-step selection approach for all the strategies by first choosing a batch of sentences and then annotating within this batch. This approach is natural for FA since the original aim is to label full sentences, and it is also commonly adopted in previous PA work (Mirroshandel and Nasr, 2011; Flannery and Mori, 2015; Li et al., 2016). Moreover, this approach makes it easier to control the reading context size for fair comparisons of different strategies as described in §3.2.

Without loss of generality, we take sequence labeling as an example and illustrate several key points in the AL process. Other tasks follow similar treatment, with details provided in Appendix A.

- **Model**. We adopt a standard BERT-based model with a CRF output layer for structured output modeling (Lafferty et al., 2001), together with the BIO tagging scheme.

- **Querying Strategy**. We utilize the query-by-uncertainty strategy with the margin-based metric, which has been shown effective in AL for structured prediction (Marcheggiani and Artières, 2014; Li et al., 2016). Specifically, each token obtains an uncertainty score with the difference between the (marginal) probabilities of the most

and second most likely label. We also tried several other strategies, such as least-confidence or max-entropy, but did not find obvious benefits.[1]

- **Sentence selection**. For both FA and PA, selecting a batch of uncertain sentences is the first querying step. We use the number of total tokens to measure batch size since sentences may have variant lengths. The sentence-level uncertainty is obtained by averaging the token-level ones. This length normalization heuristic is commonly adopted to avoid biases towards longer sentences (Hwa, 2004; Shen et al., 2018).

- **Token selection**. In PA, a subset of highly uncertain tokens is further chosen for annotation. One important question is how many tokens to select. Instead of using a pre-defined fixed selection criterion, we develop an adaptive strategy to decide the amount, as will be described in §2.2.

- **Annotation**. Sequence labeling is usually adopted for tasks involving mention extraction, where annotations are over spans rather than individual tokens. Previous work explores subsequence querying (Chaudhary et al., 2019; Radmard et al., 2021), which brings further complexities. Since we mainly explore tasks with short mention spans, we adopt a simple annotation protocol: Labeling the full spans where any inside token is queried. Note that for annotation cost measurement, we also include the extra labeled tokens in addition to the queried ones.

- **Model learning**. For FA, we adopt the standard log-likelihood as the training loss. For PA, we follow previous work (Scheffer et al., 2001; Wanvarie et al., 2011; Marcheggiani and Artières, 2014) and adopt marginalized likelihood to learn from incomplete annotations (Tsuboi et al., 2008; Greenberg et al., 2018). More details are provided in Appendix C.

## 2.2 Adaptive Partial Selection

PA adopts a second selection stage to choose highly uncertain sub-structures within the selected sentences. One crucial question here is how many sub-structures to select. Typical solutions in previous work include setting an uncertainty threshold (Tomanek and Hahn, 2009) or specifying a selection ratio (Li et al., 2016). The threshold or ratio is usually pre-defined with a fixed hyper-parameter.

This fixed selecting scheme might not be an ideal one. First, it is usually hard to specify such fixed values in practice. If too many sub-structures are selected, there will be little difference between FA and PA, whereas if too few, the annotation amount is insufficient to train good models. Moreover, this scheme is not adaptive to the model. As the model is trained with more data throughout the AL process, the informative sub-structures become less dense as the model improves. Thus, the number of selected sub-structures should be adjusted accordingly. To mitigate these shortcomings, we develop a dynamic strategy that can decide the selection in an automatic and adaptive way.

We adopt the ratio-based strategy which enables straightforward control of the selected amount. Specifically, we rank the sub-structures by the uncertainty score and choose those scoring highest by the ratio. Our decision on the selecting ratio is based on the hypothesis that a reasonable ratio should roughly correspond to the current model's error rate on all the candidates. The intuition is that incorrectly predicted sub-structures are the most informative ones that can help to correct the model's mistakes.

Since the queried instances come from the unlabeled pool without annotations, the error rate cannot be directly obtained, requiring estimation.[2] We adopt a simple one-dimensional logistic regression model for this purpose. The input to the model is the uncertainty score[3] and the output is a binary prediction of whether its prediction is confidently correct[4] or not. The estimator is trained using all the sub-structures together with their correctness on the development set[5] and then applied to the queried candidates. For each candidate sub-structure $s$, the estimator will give it a correctness probability. We

---

estimate the overall error rate as one minus the average correctness probability over all the candidates in the query set $\mathcal{Q}$ (all sub-structures in the selected sentences), and set the selection ratio $r$ as this error rate:

[1]Please refer to Appendix D.1 for more results. Note that our main focus is on AL for structured prediction, where AL selection involves not only what instances to select (acquisition function), but also at what granularity to select and annotate. In contrast with most AL work that focuses on the first aspect (and classification tasks), we mainly investigate the second one and explore better partial selection strategies. Exploring more advanced acquisition functions is mostly orthogonal to our main focus and is left to future work.

[2]Directly using uncertainty is another option, but the main trouble is that the model is not well-calibrated. We also tried model calibration by temperature scaling (Guo et al., 2017), but did not find better results.

[3]We transform the input with a logarithm, which leads to better estimation according to preliminary experiments.

[4]The specific criterion is that the $\arg\max$ prediction matches the gold one and its margin is greater than 0.5. Since neural models are usually over-confident, it is hard to decide a confidence threshold. Nevertheless, we find 0.5 a reasonable value for the ratio decision here.

[5]We re-use the development set for the task model training.

$$r = 1 - \frac{1}{n} \sum_{s \in \mathcal{Q}} p(correct = 1|s)$$

In this way, the selection ratio can be set adaptively according to the current model's capability. If the model is weak and makes many mistakes, we will have a larger ratio which can lead to more dense annotations and richer training signals. As the model is trained with more data and makes fewer errors, the ratio will be tuned down correspondingly to avoid wasting annotation budget on already-correctly-predicted sub-structures. As we will see in later experiments, this adaptive scheme is suitable for AL (§3.3).

## 2.3 Self-training

Better utilization of unlabeled data is a promising direction to further enhance model training in AL since unlabeled data are usually freely available from the unlabeled pool. In this work, we adopt self-training (Yarowsky, 1995) for this purpose.

The main idea of self-training is to enhance the model training with pseudo labels that are predicted by the current model on the unlabeled data. It has been shown effective for various NLP tasks (Yarowsky, 1995; McClosky et al., 2006; He et al., 2020; Du et al., 2021). For the training of AL models, self-training can be seamlessly incorporated. For FA, the application of self-training is no different than that in the conventional scenarios by applying the current model to all the un-annotated instances in the unlabeled pool. The more interesting case is on the partially annotated instances in the PA regime. The same motivation from the adaptive ratio scheme (§2.2) also applies here: We select the highly-uncertain sub-structures that are error-prone and the remaining un-selected parts are likely to be correctly predicted; therefore we can trust the predictions on the un-selected sub-structures and include them for training. One more enhancement to apply here is that we could further perform re-inference by incorporating the updated annotations over the selected sub-structures, which can enhance the predictions of un-annotated sub-structures through output dependencies.

In this work, we adopt a soft version of self-training through knowledge distillation (KD; Hin-

ton et al., 2015). This choice is because we want to avoid the potential negative influences of ambiguous predictions (mostly in completely unlabeled instances). One way to mitigate this is to set an uncertainty threshold and only utilize the highly-confident sub-structures. However, it is unclear how to set a proper value, similar to the scenarios in query selection. Therefore, we take the model's full output predictions as the training targets without further processing.

Specifically, our self-training objective function is the cross-entropy between the output distributions predicted by the previous model $m'$ before training and the current model $m$ being trained:

$$\mathcal{L} = -\sum_{y \in \mathcal{Y}} p_{m'}(y|x) \log p_m(y|x)$$

Several points are notable here: 1) The previous model is kept unchanged, and we can simply cache its predictions before training; 2) Over the instances that have partial annotations, the predictions should reflect these annotations by incorporating corresponding constraints at inference time; 3) For tasks with CRF based models, the output space $\mathcal{Y}$ is usually exponentially large and infeasible to explicitly enumerate; we utilize special algorithms (Wang et al., 2021) to deal with this, and more details are presented in Appendix C.

Finally, we find it beneficial to include both the pseudo labels and the real annotated gold labels for the model training. With the gold data, the original training loss is adopted, while the KD objective is utilized with the pseudo labels. We simply mix these two types of data with a ratio of 1:1 in the training process, which we find works well.

## 3 Experiments

### 3.1 Main Settings

**Tasks and data.** Our experiments[6] are conducted over four English tasks. The first two are named entity recognition (NER) and dependency parsing (DPAR), which are representative structured prediction tasks for predicting sequence and tree structures. We adopt the CoNLL-2003 English dataset (Tjong Kim Sang and De Meulder, 2003) for NER and the English Web Treebank (EWT) from Universal Dependencies v2.10 (Nivre et al., 2020) for DPAR. Moreover, we explore two more complex IE tasks: Event extraction and relation extraction.

---

[6]Our implementation is available at `https://github.com/zzsfornlp/zmsp/`.

Each task involves two pipelined sub-tasks: The first aims to extract the event trigger and/or entity mentions, and the second predicts links between these mentions as event arguments or entity relations. We utilize the ACE05 dataset (Walker et al., 2006) for these IE tasks.

**AL.** For the AL procedure, we adopt settings following conventional practices. We use the original training set as the unlabeled data pool to select instances. Unless otherwise noted, we set the AL batch size (for sentence selection) to 4K tokens, which roughly corresponds to 2% of the total pool size for most of the datasets we use. The initial seed training set and the development set are randomly sampled (with FA) using this batch size. Unless otherwise noted, we run 14 AL cycles for each experiment. In each AL cycle, we re-train our model since we find incremental updating does not perform well. Following most AL work, annotation is simulated by checking and assigning the labels from the original dataset. In FA, we annotate all the sub-structures for the selected sentences. In PA, we first decide the selection ratio and apply it to the selected sentences. We further adopt a heuristic[7] that selects the union of sentence-wise uncertain sub-structures as well as global ones since both may contain informative sub-structures. Finally, all the presented results are averaged over five runs with different random seeds.

**Model and training.** For the models, we adopt standard architectures by stacking task-specific structured predictors over pre-trained RoBERTa$_{base}$ (Liu et al., 2019) and the full models are fine-tuned at each training iteration. After obtaining new annotations in each AL cycle, we first train a model based on all the available full or partial annotations. When using self-training, we further apply this newly trained model to assign pseudo soft labels to all un-annotated instances and combine them with the existing annotations to train another model. Compared to using the old model from the last AL cycle, this strategy can give more accurate pseudo labels since the newly updated model usually performs better by learning from more annotations. For PA, pseudo soft labels are assigned to both un-selected sentences and the un-annotated sub-structures in the selected sentences.

---

[7]This heuristic will increase the actual selecting ratio, but it will only be slightly larger since there are large overlaps between sentence-wise and global highly-ranked sub-structures.

## 3.2 Comparison Scheme

Since FA and PA annotate at different granularities, we need a common cost measurement to compare their effectiveness properly. A reasonable metric is the number of the labeled sub-structures; for instance, the number of labeled tokens for sequence labeling or edges for dependency parsing. This metric is commonly adopted in previous PA work (Tomanek and Hahn, 2009; Flannery and Mori, 2015; Li et al., 2016; Radmard et al., 2021).

Nevertheless, evaluating only by sub-structures ignores a crucial hidden cost: The reading time of the contexts. For example, in sequence labeling with PA, although not every token in the sentence needs to be tagged, the annotator may still need to read the whole sentence to understand its meaning. Therefore, if performing comparisons only by the amount of annotated sub-structures, it will be unfair for the FA baseline because more contexts must be read to carry out PA.

In this work, we adopt a simple two-facet comparison scheme that considers both reading and labeling costs. We first control the reading cost by choosing the same size of contexts in the sentence selection step of each AL cycle (Line 4 in Algorithm 1). Then, we further compare by the sub-structure labeling cost, measured by the sub-structure annotation cost. If PA can roughly reach the FA performance with the same reading cost but fewer sub-structures annotated, it would be fair to say that PA can help reduce cost over FA. A better comparing scheme should evaluate against a unified estimation of the real annotation costs (Settles et al., 2008). This usually requires actual annotation exercises rather than simulations, which we leave to future work.

## 3.3 NER and DPAR

**Settings.** We compare primarily three strategies: FA, PA, and a baseline where randomly selected sentences are fully annotated (Rand). We also include a supervised result (Super.) which is obtained from a model trained with the full original training set. We measure reading cost by the total number of tokens in the selected sentences. For labeling cost, we further adopt metrics with practical considerations. In NER, lots of tokens, such as functional words, can be easily judged as the 'O' (non-entity) tag. To avoid over-estimating the costs of such easy tokens for FA, we filter tokens by their part-of-speech (POS) tags and only count the ones that

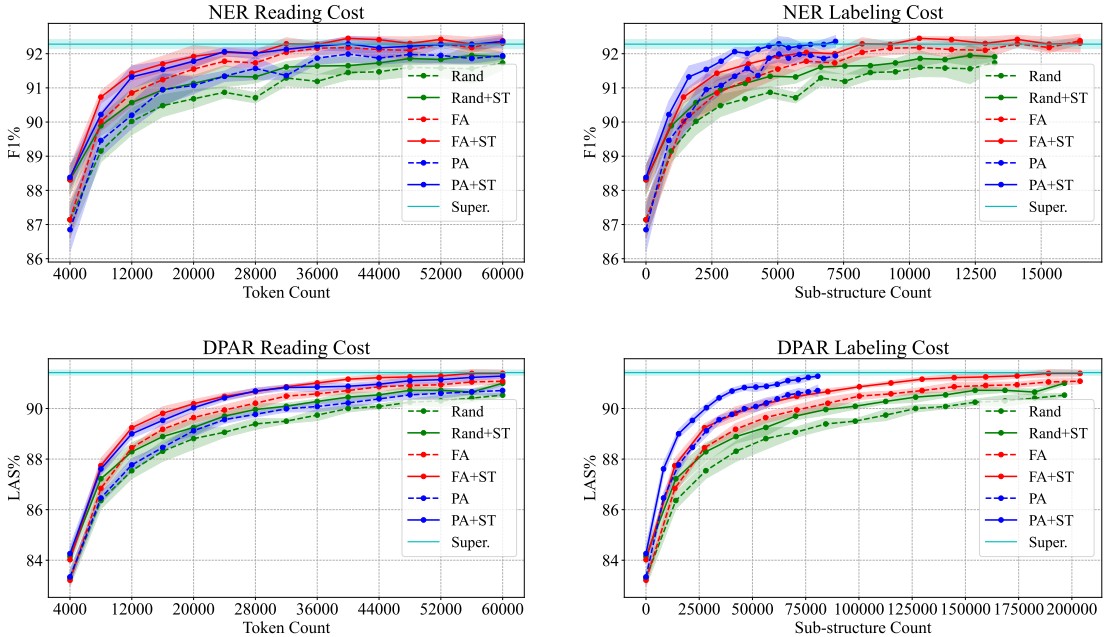

Figure 2: Comparisons according to reading and labeling cost. Each node indicates one AL cycle. For $x$-axis, reading cost (left) is measured by token numbers, while labeling cost (right) is task-specific (§3.3). NER is evaluated with labeled F1 scores on CoNLL-2003, while DPAR is with LAS scores on UD-EWT. Results are averaged over five runs with different seeds, and the shaded areas indicate standard deviations. The overall unlabeled pool contains around 200K tokens. Using AL, good performance can be obtained with less than 30% (60K) annotated.

are likely to be inside an entity mention.[8] For PA, we still count every queried token. For the task of DPAR, similarly, different dependency links can have variant annotation difficulties. We utilize the surface distance between the head and modifier of the dependency edge as the measure of labeling cost, considering that the decisions for longer dependencies are usually harder.

**Main Results.** The main test results are shown in Figure 2, where the patterns on both tasks are similar. First, AL brings clear improvements over the random baseline and can roughly reach the fully supervised performance with only a small portion of data annotated (around 18% for CoNLL-2003 and 30% for UD-EWT). Moreover, self-training (+ST) is helpful for all the strategies, boosting performance without the need for extra manual annotations. Finally, with the help of self-training, the PA strategy can roughly match the performance of FA with the same amount of reading cost (according to the left figures) while labeling fewer sub-structures (according to the right figures). This indicates that PA can help to further reduce annotation costs over the strong FA baselines.

---

[8]The POS tags are assigned by Stanza (Qi et al., 2020). For CoNLL-2003, we filter by PROPN and ADJ, which cover more than 95% of the entity tokens.

**Ratio Analysis.** We further analyze the effectiveness of our adaptive ratio scheme with DPAR as the case study. We compare the adaptive scheme to schemes with fixed ratio $r$, and the results[9] are shown in Figure 3. For the fixed-ratio schemes, if the value is too small (such as 0.1), although its improving speed is the fastest at the beginning, its performance lags behind others with the same reading contexts due to fewer sub-structures annotated. If the value is too large (such as 0.5), it grows slowly, probably because too many uninformative sub-structures are annotated. The fixed scheme with $r = 0.3$ seems a good choice; however, it is unclear how to find this sweet spot in realistic AL processes. The adaptive scheme provides a reasonable solution by automatically deciding the ratio according to the model performance.

**Error and Uncertainty Analysis.** We further analyze the error rates and uncertainties of the queried sub-structures. We still take DPAR as a case study and Figure 4 shows the results along the AL cycles in PA mode. First, though adopting a simple model, the performance predictor can give reasonable estimations for the overall error rates. Moreover, by further breaking down the error rates into selected

[9]Here, we use self-training (+ST) for all the strategies.

---

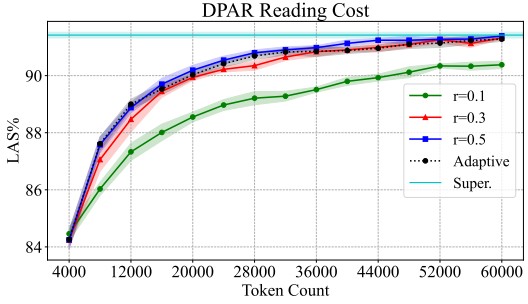

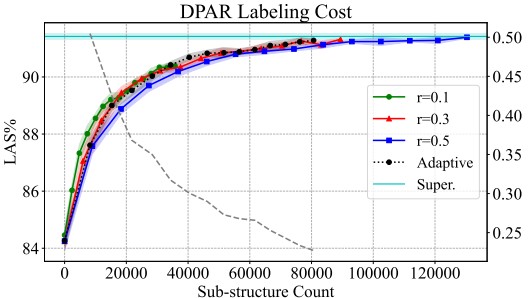

Figure 3: Comparisons of different strategies to decide the partial ratio. The first three utilize fixed ratio $r$, while "Adaptive" adopts the dynamic scheme. The grey curve (corresponding to the right $y$-axis) denotes the actual selection ratios with the adaptive scheme.

(S) and non-selected (N) groups, we can see that the selected ones contain many errors, indicating the need for manual corrections. On the other hand, the error rates on the non-selected sub-structures are much lower, verifying the effectiveness of using model-predicted pseudo labels on them in self-training. Finally, the overall margin of the selected sentences keeps increasing towards 1, indicating that there are many non-ambiguous sub-structures even in highly-uncertain sentences. The margins of the selected sub-structures are much lower, suggesting that annotating them could provide more informative signals for model training.

**Domain-transfer Experiments.** We further investigate a domain-transfer scenario: in addition to unlabeled in-domain data, we assume abundant out-of-domain annotated data and perform AL on the target domain. We adopt tweet texts as the target domain, using Broad Twitter Corpus (BTC; Derczynski et al., 2016) for NER and Tweebank (Liu et al., 2018) for DPAR. We assume we have models trained from a richly-annotated source domain and continue performing AL on the target domain. The source domains are the datasets that we utilize in our main experiments: CoNLL03 for NER and UD-EWT for DPAR. We adopt a simple model-transfer

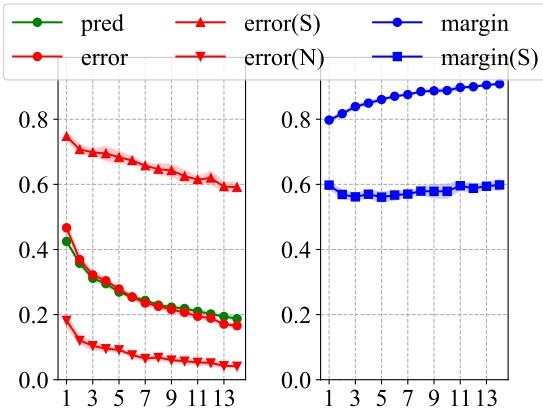

Figure 4: Analyses of error rates and uncertainties (margins) of the DPAR sub-structures in the queried sentences along the AL cycles ($x$-axis). Here, 'pred' denotes the predicted error rate, 'error' denotes the actual error rate and 'margin' denotes the uncertainty (margin) scores. For the suffixes, '(S)' indicates partially selected sub-structures, and '(N)' indicates non-selected ones. 'Margin(N)' is omitted since it is always close to 1.

approach by initializing the model from the one trained with the source data and further fine-tuning it with the target data. Since the target data size is small, we reduce the AL batch sizes for BTC and Tweebank to 2000 and 1000 tokens, respectively. The results for these experiments are shown in Figure 5. In these experiments, we also include the no-transfer results, adopting the "FA+ST" but without model transfer. For NER, without transfer learning, the results are generally worse, especially in early AL stages, where there is a small amount of annotated data to provide training signals. In these cases, knowledge learned from the source domain can provide extra information to boost the results. For DPAR, we can see even larger benefits of using transfer learning; there are still clear gaps between transfer and no-transfer strategies when the former already reaches the supervised performance. These results indicate that the benefits of AL and transfer learning can be orthogonal, and combining them can lead to promising results.

### 3.4 Information Extraction

We further explore more complex IE tasks that involve multiple types of output. Specifically, we investigate event extraction and relation extraction. We adopt a classical pipelined approach,[10] which splits the full task into two sub-tasks: the first performs mention extraction, while the second examines mention pairs and predicts relations. While

---

[10]Please refer to Appendix A for more task-specific details.

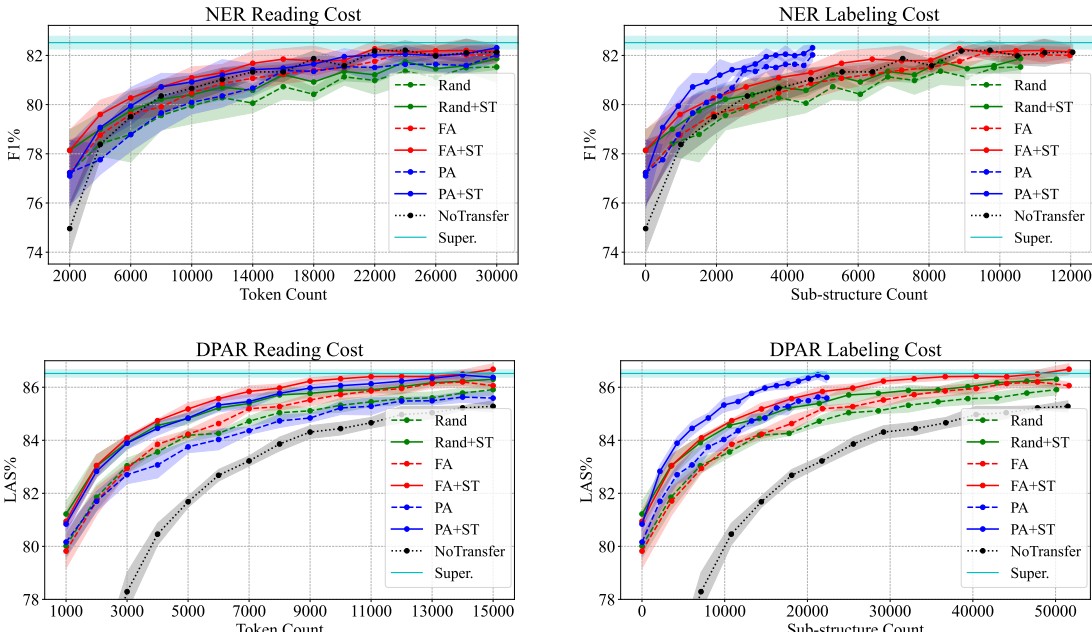

Figure 5: AL results in domain-transfer settings (CoNLL03 → BTC for NER and UD-EWT → Tweebank for DPAR). Notations are the same as in Figure 2, except that there is one more curve of "NoTransfer" denoting the setting where no transfer learning is applied (FA+ST alone).

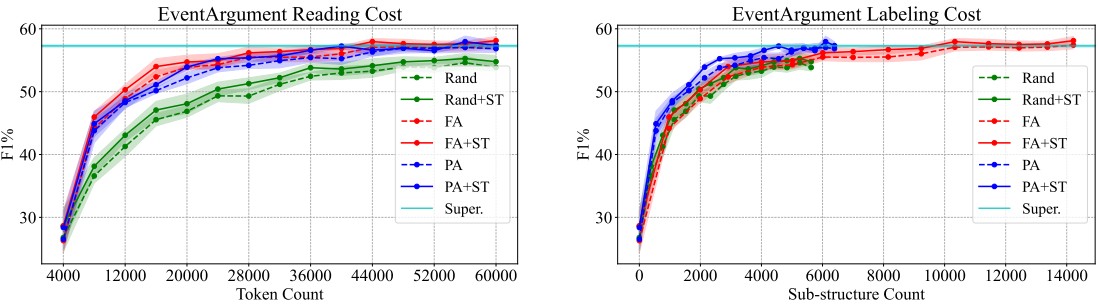

Figure 6: Results of event argument extraction on ACE05. Notations are the same as in Figure 2.

previous work investigates multi-task AL with FA (Reichart et al., 2008; Zhu et al., 2020; Rotman and Reichart, 2022), this work is the first to explore PA in this challenging setting.

We extend our PA scheme to this multi-task scenario with several modifications. First, for the sentence-selection stage, we obtain a sentence-wise uncertainty score UNC($x$) with a weighted combination of the two sub-tasks' uncertainty scores:

$$UNC(x) = \beta \cdot UNC\text{-}Mention(x) \\ + (1 - \beta) \cdot UNC\text{-}Relation(x)$$

Following Rotman and Reichart (2022), we set $\beta$ to a relatively large value (0.9), which is found to be helpful for the second relational sub-task.

Moreover, for partial selection, we separately select sub-structures for the two sub-tasks according to the adaptive selection scheme. Since the second

relational sub-task depends on the mentions extracted from the first sub-task, we utilize predicted mentions and view each feasible mention pair as a querying candidate. A special annotation protocol is adopted to deal with the incorrectly predicted mentions. For each queried relation, we first examine its mentions and perform corrections if there are mention errors that can be fixed by matching the gold ones. If neither of the two mentions can be corrected, we discard this query.

Finally, to compensate for the influences of errors in mention extraction, we adopt further heuristics of increasing the partial ratio by the estimated percentage of queries with incorrect mentions, as well as including a second annotation stage with queries over newly annotated mentions. Please refer to Appendix A.2 for more details.

We show the results of event argument extraction in Figure 6, where the overall trends are similar to

those in NER and DPAR. Here, labeling cost is simply measured as the number of candidate argument links. Overall, self-training is helpful for all AL strategies, indicating the benefits of making better use of unlabeled data. If measured by the labeling cost, PA learns the fastest and costs only around half of the annotated arguments of FA to reach the supervised result. On the other hand, PA is also competitive concerning reading cost and can generally match the FA results in later AL stages. There is still a gap between PA and FA in the earlier AL stages, which may be influenced by the errors produced by the first sub-task of mention extraction. We leave further investigations on improving early AL stages to future work. The results for the relation extraction task share similar trends and are presented in Appendix D.2, together with the results of mention extraction.

## 4 Related Work

**Self-training.** Self-training is a commonly utilized semi-supervised method to incorporate unlabeled data. It has been shown effective for a variety of NLP tasks, including word sense disambiguation (Yarowsky, 1995), parsing (McClosky et al., 2006), named entity recognition (Meng et al., 2021; Huang et al., 2021), text generation (He et al., 2020; Mehta et al., 2022) as well as natural language understanding (Du et al., 2021). Moreover, self-training can be especially helpful for low-resource scenarios, such as in few-shot learning (Vu et al., 2021; Chen et al., 2021). Self-training has also been a commonly adopted strategy to enhance active learning (Tomanek and Hahn, 2009; Majidi and Crane, 2013; Yu et al., 2022).

**PA.** Learning from incomplete annotations has been well-explored for structured prediction. For CRF models, taking the marginal likelihood as the objective function has been one of the most utilized techniques (Tsuboi et al., 2008; Täckström et al., 2013; Yang and Vozila, 2014; Greenberg et al., 2018). There are also other methods to deal with incomplete annotations, such as adopting local models (Neubig and Mori, 2010; Flannery et al., 2011), max-margin objective (Fernandes and Brefeld, 2011), learning with constraints (Ning et al., 2018, 2019; Mayhew et al., 2019) and negative sampling (Li et al., 2022).

**AL for structured prediction.** AL has been investigated for various structured prediction tasks in NLP, such as sequence labeling (Settles and Craven, 2008; Shen et al., 2018), parsing (Hwa, 2004), semantic role labeling (Wang et al., 2017; Myers and Palmer, 2021) and machine translation (Haffari et al., 2009; Zeng et al., 2019). While most previous work adopt FA, that is, annotating full structured objects for the inputs, PA can help to further reduce the annotation cost. Typical examples of PA sub-structures include tokens and subsequences for tagging (Marcheggiani and Artières, 2014; Chaudhary et al., 2019; Radmard et al., 2021), word-wise head edges for dependency parsing (Flannery and Mori, 2015; Li et al., 2016) and mention links for coreference resolution (Li et al., 2020; Espeland et al., 2020).

## 5 Conclusion

In this work, we investigate better AL strategies for structured prediction in NLP, adopting a performance estimator to automatically decide suitable ratios for partial sub-structure selection and utilizing self-training to make better use of the available unlabeled data pool. With comprehensive experiments on various tasks, we show that the combination of PA and self-training can be more data-efficient than strong full AL baselines.

## Limitations

This work has several limitations. First, the AL experiments in this work are based on simulations with existing annotations, following previous AL work. Our error estimator also requires a small development set and the proper setting of a hyperparameter. Nevertheless, we tried our best to make the settings practical and the evaluation fair, especially taking reading time into consideration. Second, in our experiments, we mainly focus on investigating how much data is needed to reach the fully-supervised results and continue the AL cycles until this happens. In practice, it may be interesting to more carefully examine the early AL stages, where most of the performance improvements happen. Finally, for the IE tasks with multiple output types, we mainly focus on the second relational sub-task and adopt a simple weighting setting to combine the uncertainties of the two sub-tasks. More explorations on the dynamic balancing of the two sub-tasks in pipelined models (Roth and Small, 2008) would be an interesting direction for future work.

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

# A More Details of Task Settings

## A.1 DPAR

- **Model**. Similar to NER, we utilize a BERT-based module to provide contextualized representations. We further stack a standard first-order non-projective graph-based parsing module based on a biaffine scorer (Dozat and Manning, 2017). The marginals for each token's head decision can be feasibly calculated by the Matrix-Tree algorithm (Koo et al., 2007; Smith and Smith, 2007; McDonald and Satta, 2007).

- **Query and Selection**. Following previous works (Flannery and Mori, 2015; Li et al., 2016), we view DPAR as a head-word finding problem and regard each token and its head decision as the sub-structure unit. In this case, the query and selection for DPAR are almost identical to the NER task because of this token-wise decision scheme. Therefore, the same AL strategies in NER can be adopted here.

- **Annotation**. In DPAR, there are no special span-based annotations as in NER; thus, we simply annotate in a word-based scheme.

- **Model learning**. Similar to NER, we adopt the log-likelihood of the gold parse tree as the training loss in FA and marginalized likelihood in PA (Li et al., 2016).

## A.2 IE

- **Tasks**. We tackle event extraction (EE) and relation extraction (RE) using a two-step pipelined approach. The first step aims to extract entity mentions for RE, and entity mentions and event triggers for EE. We adopt sequence labeling for mention extractions as in the NER task. Based on the mentions extracted in the first step, the second step examines each feasible candidate mention pair (entity pair for RE and event-entity pair for EE) and decides the relation (entity relation for RE and event argument relation for EE) for them. Since event argument links can be regarded as relations between event triggers and entities, for simplicity we will use the relational sub-task to refer to both relation and argument extraction.

- **Model**. We adopt a multi-task model similar to the one utilized in (Rotman and Reichart, 2022). With a pre-trained encoder, we take the first $N$ layers as the shared encoding module whose output representations are used for both sub-tasks. Each sub-task further adopts a private encoder that is initialized with the remaining pre-trained layers and is trained with task-specific signals. We simply set $N$ to 6, while the results are generally not sensitive to this hyper-parameter. Final task-specific predictors are further stacked upon the corresponding private encoders. We adopt a CRF layer for mention extraction and a pairwise local predictor with a biaffine scorer for relation or argument extraction.

- **Sentence selection.** For an unlabeled sentence, there is an uncertainty score for each sub-task. For mentions, the uncertainty is the average margin as in the NER task. For relations, we find that averaging uncertainties over all mention pairs has a bias towards sentences with fewer mentions. To mitigate such bias, we first aggregate an uncertainty score for each mention by taking the maximum score within all the relations that link to it and then averaging over all the mentions for sentence-level scores. Finally, the scores of the two sub-tasks are linearly combined to form the sentence-level uncertainty.

- **Partial selection.** For PA selection, the two sub-tasks are handled separately according to the adaptive ratio scheme. We further adopt two heuristics for the relational task to compensate for errors in the mention extraction. First, since there can be over-predicted mentions that lead to discarded relation queries, we adjust the PA ratio by estimating how many candidate relations contain such errors in the mentions. We again train a logistic regression model to predict whether a token is NIL (or 'O' in the BIO scheme, meaning not contained inside any gold mentions) based on its NIL probability. Then for each candidate relation, we calculate the probability that any token within its mentions is NIL. By averaging this probability of all the candidates, we obtain a rough estimation of the percentage of problematic relations, which we call it $\alpha$. Finally the PA selection ratio is adjusted by: $r_{\text{adjust}} = \alpha \cdot r_{\text{problem}} + (1 - \alpha) \cdot r_{\text{origin}}$. Here, $r_{\text{origin}}$ denotes the original selection ratio obtained from the adaptive scheme, and $r_{\text{problem}}$ denotes the selection ratio of problematic relations, which we conservatively set to 1. Secondly, since there can also be under-predicted mentions, we add a

| Data | Split | #Sent. | #Token | #Event | #Entity | #Argument | #Relation |
|------|-------|--------|--------|--------|---------|-----------|-----------|
| CoNLL03 | train | 14.0K | 203.6K | - | 23.5K | - | - |
| | dev | 3.3K | 51.4K | - | 5.9K | - | - |
| | test | 3.5K | 46.4K | - | 5.6K | - | - |
| UD-EWT | train | 12.5K | 204.6K | - | - | - | - |
| | dev | 2.0K | 25.1K | - | - | - | - |
| | test | 2.1K | 25.1K | - | - | - | - |
| ACE05 | train | 14.4K | 215.2K | 3.7K | 38.0K | 5.7K | 6.2K |
| | dev | 2.5K | 34.5K | 0.5K | 6.0K | 0.7K | 0.8K |
| | test | 4.0K | 61.5K | 1.1K | 10.8K | 1.7K | 1.7K |

Table 1: Data statistics.

second stage of querying and annotation in each AL cycle based on the annotated mentions in the first stage. This extra stage only selects relations that involve the newly added or corrected mentions. We simply reuse the selection ratio determined from the first stage and apply it to each sentence that contains such mentions. In this way, the second stage is lightweight and only requires relatively cheap re-inference for each queried sentence individually.

- **Annotation.** The annotation of the mentions is the same as in the NER task, while for the annotation of relational queries, their mentions are first examined and corrected if needed, as explained in §3.4. We measure the labeling cost by the final annotated items; thus, these extra examined mentioned will also be properly counted.

- **Model learning.** For the mention extraction subtask, the training objective is the same as in NER. For the relational sub-task, we simply adopt a local pairwise model with the standard cross-entropy loss. Since the relation model is local, no special treatment is needed for PA.

## B  Data Statistics and More Settings

**Data.** Our main experiments are conducted using the CoNLL-2003 English dataset[11] (Tjong Kim Sang and De Meulder, 2003) for NER, the English Web Treebank (EWT) from Universal Dependencies[12] v2.10 (Nivre et al., 2020) for DPAR, and English portion of ACE2005[13] (Walker et al., 2006) for IE. We utilize Stanza[14] (Qi et al., 2020) to assign POS tags for cost measurement in NER

and mention tasks. We follow Lin et al. (2020) for the pre-processing[15] of the ACE dataset. For the IE tasks on ACE, we find that the conventional test set contains only newswire documents while the training set consists of various genres (such as from conversation and web). Such mismatches between the AL pool and the final testing set are nontrivial to handle with the classical AL protocol, and we thus randomly re-split the ACE dataset (with a ratio of 7:1:2 for training, dev, and test sets, respectively). Table 1 shows data statistics. For each AL experiment, we take the original training set as the unlabeled pool, down-sample a dev set from the original dev set, and evaluate on the full test set.

**More Settings.** All of our models are based on the pre-trained $RoBERTa_{base}$ as the contextualized encoder. We further fine-tune it with the task-specific decoder in all the experiments. The number of model parameters is roughly 124M for single-output tasks and around 186M for multi-task IE tasks. For other hyper-parameter settings, we mostly follow common practices. Adam is utilized for optimization, with an initial learning rate of 1e-5 for NER and 2e-5 for DPAR and IE. The learning rate is linearly decayed to 10% of the initial value throughout the training process. The models are tuned for 10K steps with a batch size of roughly 512 tokens. We evaluate the model on the dev set every 1K steps to choose the best checkpoint. The experiments are run with one 2080Ti GPU. The training of one AL cycle usually takes only one or two hours, and the full simulation of one AL run can be finished within one day. We adopt standard evaluation metrics for the tasks: labeled F1 score for NER, labeled attachment score (LAS) for DPAR, labeled argument and relation F1 score for event arguments and relations (Lin et al., 2020).

---

[11] https://www.clips.uantwerpen.be/conll2003/ner/
[12] https://universaldependencies.org/
[13] https://catalog.ldc.upenn.edu/LDC2006T06
[14] https://stanfordnlp.github.io/stanza/

[15] http://blender.cs.illinois.edu/software/oneie/

## C Details of Algorithms

In this section, we provide more details of the algorithms for CRF-styled models (Lafferty et al., 2001). For an input instance $x$ (for example, a sentence), the model assigns a globally normalized probability to each possible output structured object $y$ (for example, a tag sequence or a parse tree) in the target space $\mathcal{Y}$:

$$p(y|x) = \frac{\exp s(y|x)}{\sum_{y' \in \mathcal{Y}} \exp s(y'|x)}$$
$$= \frac{\exp \sum_{f \in y} s(f|x)}{\sum_{y' \in \mathcal{Y}} \sum_{f' \in y'} s(f'|x)}$$

Here, $s(y|x)$ denotes the un-normalized raw scores assigned to $y$, which is further factorized into the sum of the sub-structure scores $s(f|x)$.[16] In plain likelihood training for CRF, we take the negative log-probability as the training objective:

$$\mathcal{L} = -\log p(y|x)$$
$$= -s(y|x) + \log \sum_{y' \in \mathcal{Y}} \exp s(y'|x)$$

For brevity, in the remaining, we use $\log \mathcal{Z}(x)$ to denote the second term of the log partition function. For model training, we need to calculate the gradients of the model parameters $\theta$ to the loss function. The first item is easy to deal with since it only involves one structured object, while $\log \mathcal{Z}(x)$ needs some reorganization according to the factorization:

$$\nabla_\theta \log \mathcal{Z} = \frac{\sum_{y' \in \mathcal{Y}} \exp s(y'|x) \nabla_\theta s(y'|x)}{\sum_{y'' \in \mathcal{Y}} \exp s(y''|x)}$$
$$= \sum_{y' \in \mathcal{Y}} p(y'|x) \nabla_\theta s(y'|x)$$
$$= \sum_{y' \in \mathcal{Y}} p(y'|x) \sum_{f' \in y'} \nabla_\theta s(f'|x)$$
$$= \sum_{f'} \nabla_\theta s(f'|x) \sum_{y' \in \mathcal{Y}_{f'}} p(y'|x)$$

The last step is obtained by swapping the order of the two summations, and finally, the problem is reduced to calculating each sub-structure's marginal probability $\sum_{y' \in \mathcal{Y}_{f'}} p(y'|x)$. Here, $\mathcal{Y}_{f'}$ denotes all the output structured objects that contain the sub-structure $f'$, and the marginals can usually be calculated by classical structured prediction algorithms such as forward-backward for sequence labeling (Baum et al., 1970) or Matrix-tree for non-projective dependency parsing (Koo et al., 2007; Smith and Smith, 2007; McDonald and Satta, 2007).

**Learning with incomplete annotations.** Following previous works (Tsuboi et al., 2008; Li et al., 2016; Greenberg et al., 2018), for the instances with incomplete annotations, we utilize the logarithm of the marginal likelihood as the learning objective:

$$\mathcal{L} = -\log \sum_{y \in \mathcal{Y}_C} p(y|x)$$
$$= -\log \sum_{y \in \mathcal{Y}_C} \frac{\exp s(y|x)}{\sum_{y \in \mathcal{Y}} \exp s(y|x)}$$
$$= -\log \sum_{y \in \mathcal{Y}_C} \exp s(y|x) + \log \mathcal{Z}(x)$$

Here, $\mathcal{Y}_C$ denotes the constrained set of the output objects that agree with the existing partial annotations. In this objective function, the second item is exactly the same as in standard CRF, while the first one can be calculated[17] in a modified way (Tsuboi et al., 2008).

**Knowledge distillation.** As described in the main context, we adopt the knowledge distillation objective for self-training with soft labels. For brevity, we denote the probabilities from the last model as $p'(y|x)$ and keep using $p(y|x)$ to denote the ones from the current model. Following Wang et al. (2021), the loss can be calculated by:

$$\mathcal{L} = -\sum_{y \in \mathcal{Y}} p'(y|x) \log p(y|x)$$
$$= -\sum_{y \in \mathcal{Y}} p'(y|x) s(y|x) + \log \mathcal{Z}(x)$$
$$= -\sum_{y \in \mathcal{Y}} p'(y|x) \sum_{f' \in y'} s(f'|x) + \log \mathcal{Z}(x)$$
$$= -\sum_{f'} s(f'|x) \sum_{y' \in \mathcal{Y}_{f'}} p'(y'|x) + \log \mathcal{Z}(x)$$

The loss function is broken down into two items whose gradients can be obtained by calculating marginals according to the last model or the current one, respectively.

---

[16]Such as unary and pairwise scores for sequence labeling or token-wise edge scores for dependency parsing.

[17]In our implementation, we adopt a simple method to enforce the constraints by adding negative-infinite to the scores of the impossible labels. In this case, the structures that violates the constraints will have a score of negative-infinite (and a probability of zero) and will thus be excluded.

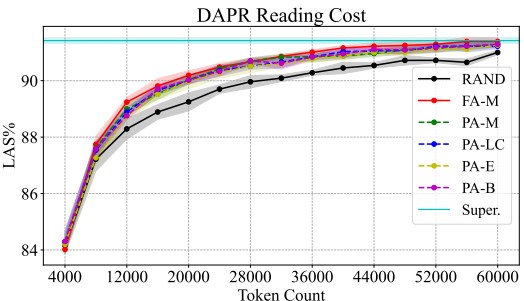 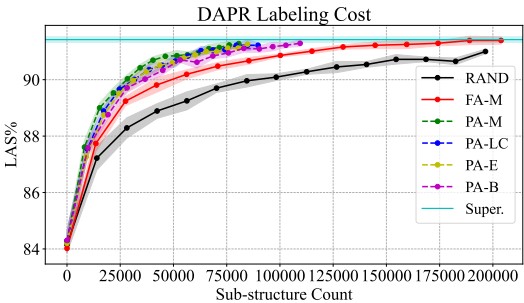

Figure 7: Comparisons of different acquisition functions for partial annotation: "-M" denotes margin-based, "-LC" denotes least-confident, "-E" denotes entropy-based, and "-B" indicates BALD.

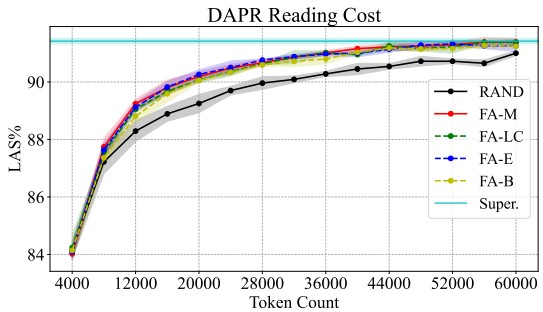

Figure 8: Comparisons of different acquisition functions for full annotation. Notations of the methods are the same as in Figure 7.

## D  Extra Results

### D.1  Using Different Acquisition Functions

In the main experiments, our acquisition function is based on margin-based uncertainty, that is, selecting the instances that have the largest marginal differences between the most and second-most confident predictions. Here, we compare it with various other acquisition functions, including least-confident (-LC), max-entropy (-E) and BALD (-B) (Houlsby et al., 2011). We take DPAR as the studying case and the results for full annotation and partial annotation are shown in Figure 8 and 7, respectively. Generally, there are no large differences between the adopted querying methods and the margin-based method can obtain the overall best results. Notice that regardless of the adopted acquisition function, we can see the effectiveness of our partial selection scheme: it requires lower labeling cost than full annotation to reach the upper bound. This shows that our method is extensible to different AL querying methods and it will be interesting to explore the combination of our method with more complex and advanced acquisition func-

tions, such as those considering representativeness.

### D.2  IE Experiments

In this section, we present more results of the IE experiments. First, Figure 9 shows the mention extraction results for the event extraction task. The overall trends are very similar to those in NER: PA can obtain similar results to FA with the same reading texts and less mention labeling cost. In Figure 10, we show the results for mention and relation extractions. In the ACE dataset, relations are very sparsely annotated, and around 97% of the entities are linked with less or equal to two relations. Considering this fact, we measure the cost of FA relation extraction by two times the annotated entities, while PA still counts the number of the queried relations. The relation results are similar to the patterns for event argument extraction, showing the benefits of selecting and annotating with partial sub-structures. Notice that in some of the mention extraction results, there seems to be less obvious differences between the AL strategies over the random baseline. This may be due to our focus on the second sub-task for relations (or event arguments), directly reflected by its high weight ($\beta$) in calculating sentence uncertainty. It will be interesting to explore better ways to enhance both sub-tasks, probably with an adaptive combination scheme (Roth and Small, 2008).

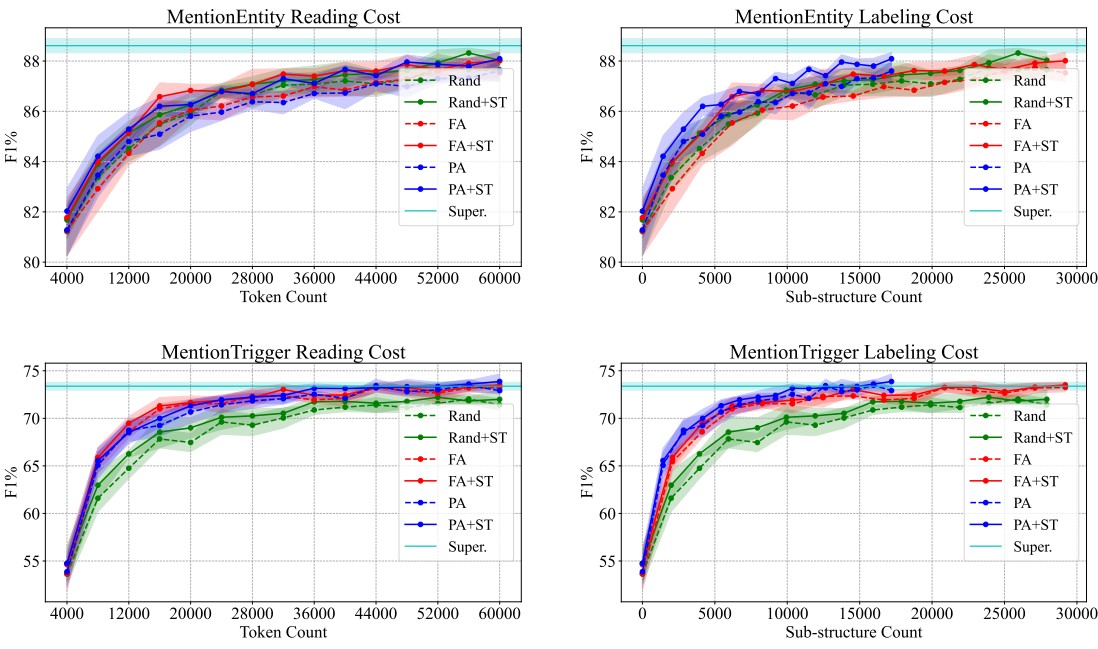

Figure 9: Results (F1) of mention extraction (entities and event triggers) for the event extraction task on ACE05 (argument results are shown in Figure 6).

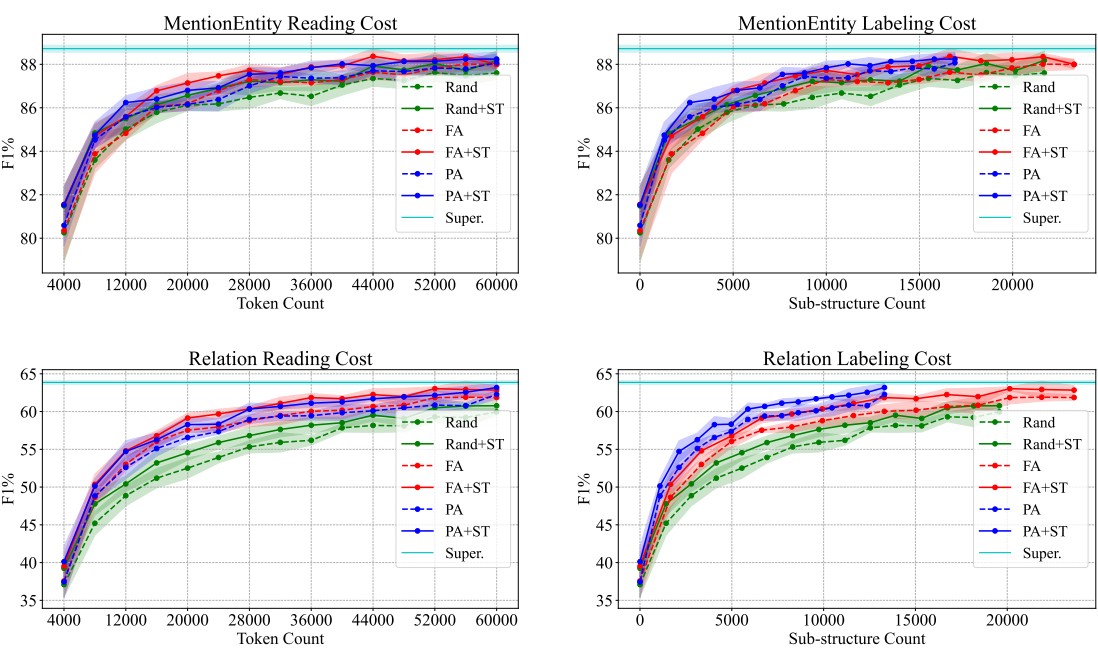

Figure 10: Results (F1) of the extraction of entity mentions and relations for the relation extraction task on ACE05.