# OpenReview forum: "Data-efficient Active Learning for Structured Prediction with Partial Annotation and Self-Training"
_EMNLP/2023/Conference — EMNLP 2023 Findings_

### Official Review · Reviewer_GDw5 · 2023-07-31

**Soundness:** 4

**Excitement:**

4: Strong: This paper deepens the understanding of some phenomenon or lowers the barriers to an existing research direction.

**Paper Topic And Main Contributions:**

This paper tackles the problem of active learning (AL) for structured prediction in NLP. The authors focus on developing a partial annotation (PA) schema with an adaptive selection ratio and combining it with self-training. The authors estimate the model’s error rate using logistic regression and then calculate the expected error rate, which they use as the ratio for a given AL step. To compare their method with the full annotation baseline (FA), the authors developed a fair evaluation considering both reading and labeling costs. While the proposed method is competitive with its FA counterpart when accounting for reading cost, it outperforms the baselines when considering labeling cost. The proposed method is thoroughly studied on several structured prediction tasks. The behavior of different aspects of the proposed method, e.g., change in adaptive ratio and the error rates throughout the AL cycle, are analyzed in detail.

**Questions For The Authors:**

A) What is a “query set” $\mathcal{Q}$? Is that the same as “sentence selection” $\mathcal{S}$ in Algorithm 1? If yes, please agree on the nomenclature and stick to it. If not, please elaborate on what is a query set since it is only mentioned once on line 242.

B) Line 257 – “We further adopt a heuristic that selects the union of sentence-wise uncertain sub-structures as well as global ones since both may contain informative sub-structures.” – Could you please elaborate on the heuristic used?

C) Line 403 - “We first control the reading cost by choosing the same size of contexts in the sentence selection step of each AL cycle (Line 4 in Algorithm 1).” - Does that mean the same sentence length? How do you ensure that different methods choose the same size of context?


**Reasons To Accept:**

- Well-written and content-dense paper.

- This relatively simple yet effective and practical idea could be helpful in real-world scenarios.

- Usually, when authors emphasize their comparison is “fair,” it immediately sounds fishy since all comparisons should be fair; otherwise, what is the point of comparing? However, the comparison authors propose does seem more suitable than the standard comparison that only considers labeling cost, and it is harsher to their proposed PA method than to the FA baselines.

- I liked the Ratio Analysis and the Error and Uncertainty Analysis paragraphs. It is always essential to do a deeper analysis of the newly proposed methods and to get an idea of how they behave and why they work.


**Reasons To Reject:**

- “Data-efficient Active Learning” in the title sounds like a pleonasm. Is it not the point of AL to be data-efficient? I believe the selling point of the paper is the adaptive ratio rather than data efficiency, and it would be better to emphasize it in the title.

- The Authors often criticize other PA approaches for reliance on hyperparameters, but their method also requires a hyperparameter. In footnote 3 the authors mention they use a confidence threshold, which is a hyperparameter, and that they set it to 0.5. The proposed method also requires a development set to train the logistic regression to estimate the error rate. Still, having a development set in practice is not often feasible. It would be good to reiterate these limitations in the Limitations section.

- I suggest commenting a bit on the results of Domain-transfer Experiments in the main part of the paper when given an extra page for the CR version.


**Reproducibility:**

4: Could mostly reproduce the results, but there may be some variation because of sample variance or minor variations in their interpretation of the protocol or method.

**Reviewer Confidence:**

4: Quite sure. I tried to check the important points carefully. It's unlikely, though conceivable, that I missed something that should affect my ratings.

---

> ### Author Rebuttal · Authors · 2023-08-28
>
> Thank you for the helpful comments!
>
> Title and domain-transfer experiments: Thanks for the suggestions! We will modify the title and move more domain-transfer results to the main content.
>
> Hyperparameter and dev set: Our motivation to add a confidence threshold is that we do not want to trust uncertainly correct ones (maybe correct by chance). This threshold is set to an intuitive value of 0.5 throughout our experiments without any tuning. For the usage of a dev set, as also discussed in our response to Reviewer 8arV, our method does not need a very large dev set since we only need to fit a simple logistic regression model. Moreover, we reuse the dev set that is used in the task model training; therefore, there will be no extra cost. We will mention these in the Limitations section.
>
> Question A: S denotes the selected sentences and Q denotes all the sub-structures in S. We will add more details for this.
>
> Question B: Assuming we have a selection ratio, for example, 0.3, we can rank all the sub-structures globally and take the first 30 percent, or, we can rank each sentence's sub-structures individually and select 30 percent within each sentence. We find that combining both leads to more robust results and thus simply take the union.
>
> Question C: This is measured by the total number of tokens (total sentence length), as denoted by the x-axis of all the “Reading Cost” figures. For example, in Figure 2, we select sentences whose total length is around 4K in each AL cycle. In the sentence selection stage, we rank all the sentences according to their sentence-level uncertainty (as discussed starting from Line 159) and select until hitting the 4K budget.

---

### Official Review · Reviewer_gpGM · 2023-08-05

**Soundness:** 3

**Excitement:**

3: Ambivalent: It has merits (e.g., it reports state-of-the-art results, the idea is nice), but there are key weaknesses (e.g., it describes incremental work), and it can significantly benefit from another round of revision. However, I won't object to accepting it if my co-reviewers champion it.

**Paper Topic And Main Contributions:**

This paper provides a framework to combining partial annotation with self-training to reduce annotation cost for structured prediction using active learning.

This paper addressed the annotation for structured prediction and to reduce annotation cost, within the framework of active learning, the developed Adaptive Partial Selection could be more flexible for diverse structures.

**Reasons To Accept:**

1. The motivation of this work is very clear and meaningful.
2. The proposed Adaptive Partial Selection provides the idea for addressing real annotation difficulties.
3. The writing is well-organised.



**Reasons To Reject:**

Fail to provide solid support for the design details. For example, in Adaptive Partial Selection, the assumption is based on the intuition that incorrectly predicted sub-structures are the most informative ones that can help to correct the model’s mistakes. Also, for optimizing self training objective, regarding the derivative of partition function with respect to \theta, is there any explanation for the transformation to the restricted sub-structure?



**Reproducibility:**

4: Could mostly reproduce the results, but there may be some variation because of sample variance or minor variations in their interpretation of the protocol or method.

**Reviewer Confidence:**

3: Pretty sure, but there's a chance I missed something. Although I have a good feel for this area in general, I did not carefully check the paper's details, e.g., the math, experimental design, or novelty.

---

> ### Author Rebuttal · Authors · 2023-08-28
>
> Thank you for the helpful comments!
>
> Adaptive Partial Selection: We think that our analyses in Figures 3 and 9 provide empirical evidence for the effectiveness of our scheme: the adaptive selection scheme is the overall best performing. Moreover, another way to view our motivation is that the ideal goal of the annotation is to obtain fully annotated instances, in this way, if the prediction is incorrect, we should select and correct the corresponding sub-structure.
>
> Derivative of the partition function with restricted sub-structures: In the case that some of the sub-structures are restricted to pre-determined labels, we can still adopt dynamic-programming styled algorithms as in the normal scenario, but force the labels for the restricted sub-structures. Actually, in our implementation, we adopt a simple method to enforce this by adding -inf to the scores of the impossible labels. In this case, the Y that violates the constraints will have a score of -inf (a probability of 0) and will thus be excluded. We will include more details on this in later versions.
>
> Please let us know if there are any other places where you think there are needs of more explanations.

---

### Official Review · Reviewer_8arV · 2023-08-06

**Soundness:** 3

**Excitement:**

3: Ambivalent: It has merits (e.g., it reports state-of-the-art results, the idea is nice), but there are key weaknesses (e.g., it describes incremental work), and it can significantly benefit from another round of revision. However, I won't object to accepting it if my co-reviewers champion it.

**Paper Topic And Main Contributions:**

The paper tackles active learning with partial annotation problem for structured prediction. It introduces a method that adaptively determines the selection ratio for sub-structures to annotate. Coupled with self-training, the proposed method provides improvement over existing full-annotation baselines.

**Questions For The Authors:**

See above.

**Reasons To Accept:**

- The paper is clear and easy to read.
- The training of a calibration model, from uncertainty to correctness, to adaptively determine the selection ratio is interesting.
- I appreciate the inclusion of reading cost and labeling cost to evaluate the methods from more angles.

**Reasons To Reject:**

- The main contribution of the work is the proposed method that trains the error estimator to determine the selection ratio for sub-structure. While the idea is interesting, it is quite straightforward. My main concern for this method is the requirement for a validation set in order to train the error-estimator. It is not clear how the performance is sensitive to the size of the validation set.
- There also doesn't seem to have a strong motivation for the proposed method. For instance, why do we need to use the estimated error rate, instead of simply using the current uncertainty? Is there any theoretical reason for this or empirical evidence to support?
- The self-training scheme is more of a direct application here. It's great that the authors explore this direction in the paper but the contribution here is relatively limited.


**Reproducibility:**

4: Could mostly reproduce the results, but there may be some variation because of sample variance or minor variations in their interpretation of the protocol or method.

**Reviewer Confidence:**

3: Pretty sure, but there's a chance I missed something. Although I have a good feel for this area in general, I did not carefully check the paper's details, e.g., the math, experimental design, or novelty.

---

> ### Author Rebuttal · Authors · 2023-08-28
>
> Thank you for the helpful comments!
>
> Validation Set: Our method does not require a large validation set to obtain reasonable estimations, since the estimator is a simple logistic regression model and does not need much data to fit. Notice that we utilize the same dev set for both the training of our main model and the estimator; in this way, there is no extra cost.
>
> Estimated error rate: Our motivation for using error rate is that: if a sub-structure is wrongly predicted, we should pick it for annotation; whereas if the model prediction is correct, then there is no need to select. In this way, the ideal ratio should be the error rate if we want all the sub-structures' labels to be correct. Uncertainty can be an indicator, but the main trouble is that the model is not calibrated. Most of the predictions seem to be confident (for example, margin > 0.9) according to the model's output, but many can be wrong. This will also lead to a small selection ratio and as Figures 3 and 9 show, the model will be under-trained if there are not enough training instances.
>
> Self-training: We agree that using self-training is not a novel idea as we have already acknowledged (for example, Line 98). Nevertheless, we think its application in the context of partial AL is very natural and motivates our adaptive ratio scheme (annotate the wrongly predicted ones and trust the correct ones). We think the combination of AL and self-training with this adaptive ratio scheme is a novel idea.

---

### Meta-Review · Area_Chair_aFe3 · 2023-09-19

**Recommendation:** 3

**Metareview:**

The paper suggests partial annotation coupled with self-training for active learning of structured prediction. The paper is clearly-written, and ideas are interesting, addressing issues in real annotations. However, in the term of self-training, the contribution is limited. In addition, the design details should be provided with more explainations.

---

### Decision · Program_Chairs · 2023-10-07

**Decision:**

Accept-Findings

**Comment:**

The paper suggests partial annotation coupled with self-training for active learning of structured prediction. The paper is clearly-written, and ideas are interesting, addressing issues in real annotations. However, in the term of self-training, the contribution is limited. In addition, the design details should be provided with more explainations.